# When Cultural Resources Amplify Psychological Strain: Off-Work Music Listening, Homophily, and the Homesickness–Burnout Link Among Migrant Workers

**DOI:** 10.3390/bs15050666

**Published:** 2025-05-13

**Authors:** Chenyuan Gu, Zhuang Ma, Xiaoying Li, Jianjun Zhang, Qihai Huang

**Affiliations:** 1School of Music and Dance, Communication University of Zhejiang, Hangzhou 310018, China; gucheny@cuz.edu.cn; 2Huddersfield Business School, University of Huddersfield, Huddersfield HD1 3DH, UK; 3Department of Strategy, Operations and Entrepreneurship, University of Essex, Essex CO4 3SQ, UK; 4Guanghua School of Management, Peking University, Beijing 100871, China

**Keywords:** homesickness, music listening, migrant, homophily, job resources and demands, burnout

## Abstract

Hundreds of millions of migrants experience frequent homesickness that affects their psychological wellbeing. This study integrates the job-demands–resources model and temporal comparison theory to examine how music listening and similar activities involving coworker homophily and roommate homophily influence the relationship between homesickness and burnout. Our analysis of survey data from 2493 migrant workers reveals that off-work music listening strengthens the positive relationship between homesickness and burnout. Furthermore, coworker homophily and roommate homophily enhance the strength of the interaction between off-work music listening and homesickness as a predictor of burnout. Our findings demonstrate how seemingly supportive job resources can transform into psychological demands and thus have important theoretical and managerial implications.

## 1. Introduction

Globally, the number of international migrants reached 281 million (3.6% of the population) in mid-2020 and rose to 304 million by 2024 (United Nations Migration Data Portal). China is home to the largest domestic labour migration, with 297.5 million rural workers engaged in non-agricultural employment outside their hometowns as of 2023 ([114]). In this study, we define migrants broadly as individuals who relocate geographically from their place of origin, while the term “migrant workers” specifically refers to those who move primarily for employment opportunities ([41]). In China, rural-to-urban migrant workers (nongmingong) constitute a distinctive social category of people who relocate from rural areas to urban centres in search of better economic opportunities while maintaining household registration (hukou) in their places of origin ([14]). These workers typically experience profound psychological challenges, particularly homesickness, which is a preoccupation with home, family, and friends ([22]; [92]). As a stressor ([56]; [64]; [96]), homesickness can negatively affect psychological wellbeing ([2]; [15]).

Burnout among migrant workers is prevalent ([53]; [111]; [119]). Financial pressures, which drive them to overwork, may lead to burnout ([55]; [118]). According to the job-demand–resources (JD–R) model ([4]), specific risk factors associated with job stress or burnout can be classified as job demands or job resources. Job demands refer to the physical, social, or organisational aspects of the job that require sustained physical or mental effort ([79]). We conceptualise homesickness as a job demand because managing feelings of separation and longing requires significant emotional and cognitive resources that are then unavailable for work tasks. Homesickness imposes an affective–cognitive burden that migrant workers must regulate continuously while performing their jobs ([20]). By contrast, the physical, psychological, social, and organisational aspects of the job that help in achieving work goals, reducing job demands, and stimulating personal growth and development are job resources. A key question arises: Does homesickness, as a significant emotional job demand, deplete migrant workers’ psychological resources and result in burnout, and if so, how? [92] ([92]) speculate on such a relationship without providing any empirical evidence.

The JD–R model posits that resources such as social networks and leisure activities such as music listening serve as social and cultural resources that buffer against job demands ([19]; [45]; [74]; [75]) such as homesickness. However, temporal comparison theory suggests these factors might trigger comparisons between the present (i.e., the current place) and past circumstances (i.e., the hometown), potentially transforming supportive resources into additional sources of strain ([1]; [109]; [110]). For instance, music tied to cultural identity might replenish emotional resources by fostering hometown connections ([52]; [102]) yet simultaneously evoke ruminative thoughts about displacement ([18]). May listening to music, considered a resource that buffers against burnout, alternatively make homesick migrants more vulnerable to burnout?

Furthermore, homophily in social networks ([68]), i.e., the tendency to associate with similar others, suggests that migrant workers are most likely to attend cultural activities with people who share similar backgrounds. While homophily in workplace and living environments can influence wellbeing and performance ([68]; [74]) and music listening can affect employees’ psychological states ([58]), it remains unclear how music listening combined with homophily interactively influence the homesickness–burnout link among migrant workers. Addressing the above gaps and questions is crucial, as this information can be used to mitigate the substantial productivity losses and turnover costs associated with burnout-related withdrawal behaviours.

By integrating two theoretical lenses, the job-demands–resources (JD–R) model ([19]; [45]; [74]; [75]) and temporal comparison theory ([1]; [109]; [110]), we examine how social homophily and off-work music listening shape the relationship between homesickness and burnout among migrant workers in China. Like many other emerging economies, China has witnessed large-scale migration from rural to urban areas ([76]). Among China’s 297.5 million migrant workers ([71]), the majority migrate from rural to urban areas, creating significant geographical and cultural distance from their homes ([112]). The psychological strain of separation from family and familiar surroundings imposes an emotional burden. These workers typically live in dormitories or shared accommodations and work in labour-intensive industries, and their coworkers and roommates are essential members of their social networks ([97]). We differentiate between roommate and coworker homophily to account for the distinct social contexts in which these relationships function. Roommate homophily, rooted in informal residential settings, may facilitate collective reminiscence and emotional sharing during off-work hours, while coworker homophily, situated in formal workplace environments, may influence interactions during off-work time ([35]; [49]). By examining these two contexts separately, we aim to uncover how the social dynamics of living and working environments shape the psychological effects of homophily and music listening.

Our focus on off-work music listening and informal social relationships enriches the current burnout research literature, which has primarily examined formal workplace dynamics and individual coping strategies. We address the growing critiques of psychological reductionism in burnout research ([9]), which call for greater attention to the sociocultural contexts in which psychological strain emerges. By examining how cultural practices and social relationships outside the workplace influence burnout, we acknowledge that employees’ wellbeing cannot be reduced to workplace factors alone but must be understood within broader patterns of daily life and social embeddedness. This cultural–sociological approach addresses emerging perspectives in music and health research that recognise music’s ambivalent psychological effects ([12]; [86]), extending these insights by demonstrating how social context shapes music’s impact on wellbeing. Our cultural–sociological perspective on emotional life under precarious work conditions offers a complementary picture of how migrant workers experience and manage the psychological demands of displacement.

This study makes several important contributions. First, we frame homesickness as a significant job demand by demonstrating its direct relationship with workplace burnout, extending previous research that has primarily focused on general psychological distress ([22]; [56]). Second, by integrating the JD–R model and temporal comparison theory, we advance our understanding of the contingent nature of cultural and social resources. Specifically, we reveal how off-work music listening, traditionally viewed as a cultural resource, can interact with social homophily to amplify rather than mitigate the relationship between homesickness and burnout. This challenges the prevailing assumption about resources having universally beneficial effects, demonstrating that personal and social resources can paradoxically transform into demands when they trigger upward temporal comparisons between current circumstances and idealized memories of home. Third, we extend research on social relationships among migrant workers by examining how homophily can transform potential support mechanisms into sources of strain. This advances previous work, which has primarily focused on the presence or absence of social relationships without considering their homophilic nature.

## 2. Theoretical Background and Hypothesis Development

### 2.1. Homesickness and Burnout

According to the JD–R model, homesickness represents a significant emotional job demand that can deplete workers’ psychological resources and result in burnout ([2]; [19]). Homesickness is particularly salient for migrant workers, who leave their hometowns, families, and established social networks to work in unfamiliar environments. Homesick migrant workers may be preoccupied with thoughts of home, feelings of loss due to separation from family and friends, and a persistent longing to return home ([92]). Such feelings can be particularly intense as migrant workers navigate work responsibilities and adapt to a new social environment ([57]; [104]). The cognitive and emotional resources required to cope with homesickness can drain the psychological energy needed for work tasks and emotional regulation ([39]; [91]). For instance, migrant workers may struggle to concentrate on tasks while managing intrusive thoughts about home or may experience difficulty maintaining emotional composure during customer interactions due to underlying feelings of displacement ([26]; [32]).

While the JD–R model identifies the resource-depleting nature of homesickness as a job demand ([17]), temporal comparison theory can help unravel the psychological mechanisms underlying this depletion ([1]). When experiencing homesickness, migrant workers make temporal comparisons between their current circumstances and their past life at home ([85]). These comparisons, intensified by China’s cultural emphasis on family bonds and filial piety, can transform traditional resources into sources of additional strain ([115]). While shared hometown identity initially facilitates social support, frequent interactions with coworkers from the same hometown may trigger collective reminiscence about home life, thereby intensifying feelings of current displacement ([60]).

According to the JD–R model, when job demands such as homesickness gradually exceed a migrant worker’s adaptive resources, burnout may emerge ([4]). Recent extensions of the JD–R model have further emphasized its contextual sensitivity, particularly in terms of how cultural and social factors can transform the functionality of resources ([5]). Burnout develops as workers deplete their psychological resources to manage work demands and the emotional strain of separation from home ([117]). This process is particularly evident in labour-intensive industries, where migrant workers take on physically demanding tasks while managing psychological adaptation ([57]; [63]).

Studies have consistently associated homesickness with negative psychological outcomes in individuals who are separated from home. For instance, international students experiencing homesickness report higher levels of psychological distress ([78]; [90]), while boarding-school students with homesickness demonstrate decreased psychological adjustment ([23]; [99]). Recent studies have confirmed the relationship between emotional demands and burnout across cultural contexts, with particularly strong effects observed in displaced populations ([81]; [82]). Moreover, homesickness contributes to emotional drainage and reduced work engagement ([3]; [89]). Studies of rural-to-urban migrant workers have revealed that this population experiences significant acculturative stress from separation from home and family, which can manifest in psychological strain ([62]; [121]). The impact appears particularly pronounced in manufacturing settings, where migrant workers face repetitive tasks and limited opportunities for meaningful social interaction ([100]). In short, prolonged separation from familiar environments, emotional strain from displacement, and limited resources for coping can result in burnout. Therefore, we propose the following:

**Hypothesis 1 (H1).** 
*Homesickness is positively related to burnout among migrant workers.*


### 2.2. Music Listening as a Resource-Replenishing or Demand-Amplifying Activity

Current neuroscience research has found that musical features interact with individual memory systems and cultural schemas to produce complex affective responses ([25]; [44]). From a JD–R perspective, music listening serves as a personal resource that helps regulate emotions, reduce stress, and buffer against job demands ([4]; [88]). Workers may utilize music as a coping mechanism to manage workplace stress and restore depleted psychological resources ([31]; [54]). Similarly, for migrant workers, music can serve as an accessible means to replenish emotional resources and maintain connections to cultural identity, similarly to how international students use cultural media to cope with adjustment challenges ([103]). As an example, folk music can evoke memories of hometowns, traditional festivals, or family gatherings, reminding migrant workers of their cultural roots. This can foster a sense of comfort and continuity ([10]). Music can function as a positive resource. However, music can also become a trigger for negative emotions, depending on context ([12]; [86]). This ambivalence is particularly relevant for migrant populations, for whom culturally familiar music may simultaneously provide comfort and heighten awareness of cultural displacement ([47]), with the latter intensifying the experiences of displacement ([42]; [65]).

Temporal comparison theory suggests that music listening may also amplify demands by triggering upward comparisons between migrants’ current circumstances and idealized memories of home ([1]; [109]). Research on music in diasporic communities shows how musical engagement creates complex emotional responses among displaced populations ([77]; [80]). [13] ([13]) discussed reflective and restorative nostalgia. Restorative nostalgia’s idealisation of homeland may intensify migrants’ feelings of displacement. For migrant workers, music might transform from a cultural resource to a psychological demand by activating these nostalgic orientations and triggering temporal comparisons between an idealized past and the present reality. This is particularly likely to occur during off-duty time, when workers have greater cognitive bandwidth for reflection and emotional processing ([93]; [95]) and the absence of work-related distractions allows for more pronounced engagement with nostalgic or idealized memories ([108]; [122]).

Music listening can transform from a resource into a demand through several mechanisms, particularly during off-work time. First, music can intensify existing emotional states rather than merely altering them ([29]). For homesick individuals, this amplification can strengthen negative effects associated with separation, such as loneliness ([61]). Research on nostalgia-induced music listening reveals that while familiar music can evoke positive emotions initially, it often leads to heightened awareness of temporal and geographical displacement and thus results in mixed emotional outcomes ([6]; [46]).

Second, music is a potent trigger for autobiographical memories, often evoking vivid recollections of past experiences ([43]; [87]). Such memories may remind migrant workers of past social connections and present isolation, intensifying feelings of displacement and reinforcing their internal identity as migrants ([8]; [43]). Studies on consumption of cultural media among displaced populations show that autobiographical memories triggered by culturally familiar music can exacerbate homesickness by reinforcing the perceived loss of social and cultural ties ([28]). Music can also evoke nostalgia, which further amplifies emotional responses, especially when individuals reflect on idealized memories of home ([108]). Emotional responses to music can intensify during leisure or off-work time, as greater cognitive capacity facilitates deeper reflection and emotional processing ([27]).

Third, exposure to cultural elements such as music can raise awareness of cultural identity and cultural distance ([11]). For migrant workers, this increased awareness of cultural identity may amplify feelings of disconnection from their hometown culture ([10]). Research on cultural adaptation suggests that heightened salience of cultural identity can increase psychological strain as individuals become more acutely aware of their cultural dislocation ([7]; [121]). This strain is further exacerbated when individuals perceive a lack of cultural fit or belonging in their new environment ([72]).

In summary, music listening during off-work time can transform from a potential resource into a demand by intensifying awareness of social losses through activation of autobiographical memories ([43]), amplifying existing homesickness through emotional intensification ([29]), heightening cultural identity threats through increased awareness of cultural distance ([7]), and exacerbating feelings of isolation by contrasting solitary urban experiences with communal rural practices ([10]). Therefore, we offer the following hypothesis:

**Hypothesis 2 (H2).** 
*Music listening during off-work time enhances the strength of the positive relationship between homesickness and burnout among migrant workers.*


### 2.3. Music Listening with Homophilic Coworkers and Roommates

According to [68] ([68]), individuals are naturally drawn to forming stronger relationships with those who share similar characteristics, attitudes, and backgrounds, i.e., those who exhibit homophily. Recent studies demonstrated how digital technologies have transformed but not eliminated homophilic preferences among migrant communities ([67]; [106]). Within the JD–R model, homophily with coworkers and roommates represents a potential social resource that can provide instrumental and emotional support, helping migrant workers manage job demands ([4]; [37]; [75]; [107]). However, the effects of homophily are context-dependent, with coworker and roommate homophily operating in distinct social settings that shape their psychological impacts.

In workplace settings, coworker homophily (i.e., shared characteristics such as motivations for migration, work attitudes, and cultural values) can facilitate both supportive and potentially problematic interactions. Homophily functions not only as a social sorting mechanism but also as a means of boundary maintenance and identity reinforcement. Research on elite social networks has demonstrated how homophilous relationships serve to reinforce in-group boundaries and create subjective distinction from out-groups ([16]). While they are typically studied in privileged populations, these boundary-drawing functions of homophily are equally relevant to understanding migrant workers’ experiences.

Shared migration motivations, such as the pursuit of better economic opportunities or the desire to support family back home, create strong bonds among migrant workers ([21]). Similarly, shared work attitudes and future aspirations, such as goals related to skill development or entrepreneurial ambitions, foster close relationships, particularly when workers share similar trajectories ([106]). Migrant workers’ shared values, which originate from hometown cultures and family obligations, further enhance these connections, providing a sense of mutual understanding ([98]; [105]).

As we argued earlier, music can trigger migrant workers’ autobiographical memories, intensifying the salience of their identities and heightening awareness of displacement ([8]; [43]). The tendency towards socially embedded coping with homesickness may result in social interaction in homophilic environments, as mentioned above, which may in turn facilitate co-rumination, i.e., the excessive discussion of problems within close relationships that paradoxically intensifies negative emotions ([33]; [84]), and emotional contagion, whereby emotional states spread among socially connected individuals ([36]; [83]). Studies have confirmed cultural variations in co-rumination patterns, with collectivist norms intensifying the emotional impact of problem-focused conversations ([30]; [51]). For migrant workers, listening to music evokes memories of home due to internal identity processing ([8]; [43]). Doing so with culturally similar coworkers may trigger collective temporal comparisons between their present circumstances and past experiences at home. These comparisons, particularly when they emphasize losses or unmet aspirations, can transform potential social supports into sources of shared distress. For instance, workers from neighbouring hometowns may share similar festival music or folk songs, making these musical triggers more universally meaningful within the group. Similarities in age and life stage may lead migrant workers to recall comparable experiences, potentially intensifying the contrast with current circumstances ([108]).

During off-work time, the combination of shared work attitudes, migration motivations, and cultural values creates a potent context for collective reminiscence. When listening to music with colleagues sharing substantial similarities, such as backgrounds, values, and future plans, they tend to have deeper discussions about the challenges of their migration journey. The alignment of their work attitudes and migration motivations might prompt them to collectively question whether their current situations align with their shared aspirations, especially when music evokes thoughts of their homes ([29]). This communal reflection can enhance feelings of dissatisfaction or longing, especially when the music evokes nostalgic or idealized recollections of their past lives. Therefore, we offer the following hypothesis:

**Hypothesis 3a (H3a).** 
*Coworker homophily strengthens the moderating effect of non-working-time music listening on the homesickness–burnout relationship, such that the positive relationship between homesickness and burnout becomes stronger when listening to music with high-homophily coworkers.*


In residential settings, roommate homophily (i.e., similarity in traits such as age, migration experiences, and geographical proximity of hometowns) can significantly influence migrant workers’ psychological adjustment. Younger workers, for example, often face similar life-stage challenges in urban environments, such as navigating career development or adapting to urban lifestyles, which can foster solidarity and mutual support ([120]). Shared migration experiences, such as leaving home and adapting to urban life, can create powerful bonds through mutual understanding ([113]). Moreover, migrant workers with hometowns in geographical proximity may share similar dialects and common customs that lead to closer relationships ([40]; [116]).

While these shared traits can provide emotional comfort, they can also intensify feelings of homesickness. [92] ([92]) found that homesick individuals often engage in collective reminiscence about home with others from similar backgrounds, which can heighten awareness of separation and cultural dislocation. During off-work time, music listening may result in comparisons about disconnected social ties and cultural celebrations ([108]; [122]). For instance, migrants sharing living spaces who come from proximate areas may recognize common music styles or traditional songs, thus enhancing their collective resonance of such audio cues. The generational commonalities among roommates can result in recollection of musical experiences from their respective hometowns, thus intensifying the contrast with their current circumstances ([29]).

When roommates who share multiple dimensions of homophily interact through music, their similar values and migration experiences may lead to deeper discussions about the sacrifices of their migration journeys. The alignment of their cultural backgrounds and shared memories may prompt them to reflect collectively on their separation from home, especially when musical selections remind them of their origins. This process can operate through emotional contagion, i.e., shared experience of listening to nostalgic music that amplifies feelings of homesickness ([83]), and temporal comparison, i.e., contrasting current urban circumstances with idealized memories of home ([1]). Therefore, we offer the following hypothesis:

**Hypothesis 3b (H3b).** 
*Roommate homophily strengthens the moderating effect of non-working-time music listening on the homesickness–burnout relationship, such that the positive relationship between homesickness and burnout becomes stronger when listening to music with roommates with whom one shares high homophily.*


Figure 1 presents the conceptual model.

## 3. Methods

### 3.1. Sampling

Our empirical context is Guangdong Province, China, which hosts the country’s largest population of migrant workers (42.2 million, accounting for 14.4% of China’s total of 292.5 million migrant workers in 2021; [70]). Guangdong’s rapid economic development and substantial migrant-worker population provide an ideal setting for studying migrant workers.

We employed a stratified random sampling approach. First, we obtained a list of 1000 firms from local governments across five major cities[note 1]. In Guangdong, we randomly selected 180 firms, with the number of firms from each city proportionate to the size of its migrant worker population. Of these, 158 firms agreed to participate (87.8% response rate). The participating firms were predominantly from the manufacturing sector, with foreign-owned firms (57.90%) and privately owned firms (30.95%) firms making up the majority of our sample.

Questionnaires were distributed through the HR departments of participating firms. To protect respondents’ privacy and encourage honesty in responses, we ensured confidentiality and anonymity for all participants. From 4076 returned questionnaires, we obtained 2493 valid responses from 122 firms after excluding incomplete or invalid responses (61.2% valid response rate). To address potential common-method bias, we conducted Harman’s single-factor analysis, which revealed no significant concerns.

The sample represents Guangdong’s migrant-worker population and has a balanced gender distribution (46.33% male, 53.67% female).

### 3.2. Measures

Burnout was measured using three specific items adapted from [119] ([119]), which are similar to those used by [66] ([66]). Respondents were asked to rate on a scale of 1 to 5 how well three statements describe their experience over the past two weeks. One example is, “I feel emotionally drained from my work”.

Homesickness was measured using four items adapted by [94] ([94]). Respondents were asked to rate on a scale of 1 to 5 how well the four statements fit their situations. One example is, “I often miss the life back in my hometown”.

Off-work music listening was a dummy variable that equalled 1 if an individual listened to music during non-working time and 0 if they did not.

Homophily with coworkers and roommates was measured using seven items adapted from [68] ([68]). This scale captures multiple dimensions of similarity that may influence migrant workers’ social relationships. For coworker homophily, we measured similarity in motivations for migration (i.e., ‘They came here to work for similar reasons as I did’), work attitudes (i.e., ‘They have similar attitudes toward work as I do’), and cultural values (i.e., ‘They share similar values and beliefs with me’). For roommate homophily, we assessed similarity in age group (i.e., ‘They are in a similar age group as I am’), migration experiences (i.e., ‘They have similar experiences of leaving home as I do’), and geographical proximity of hometowns (i.e., ‘Their hometowns are close to mine’).

We included several demographic and background characteristics as control variables. Gender (male = 1) showed no significant effect in any model (β = 0.0207 to 0.0352, *p* > 0.10), suggesting no systematic differences in burnout between male and female workers. Age showed a consistent negative relationship with burnout across all specifications (β ≈ −0.018, *p* < 0.01), indicating that older workers generally reported lower levels of burnout.

Marital status (married = 1) and education level showed no statistically significant associations with burnout (β = −0.037 to −0.097 and β = −0.045 to −0.061, respectively, *p* > 0.10). However, education showed marginal significance in models with coworker homophily (*p* < 0.10). Job tenure, measured in years, had no significant effect on burnout in any model (β ≈ −0.0004 to −0.005, *p* > 0.10).

Additionally, we controlled for migrant workers’ cross-province migration status and positive association with burnout in models including coworker homophily (β ≈ 0.30, *p* < 0.05), which was significant in most specifications.

Table 1 presents the descriptive statistics. Around 45% of the respondents were males. The average was 25, with a standard deviation of 6. In total, 76.9% of the respondents were cross-province migrants. The correlation between coworker homophily and roommate homophily is 0.697, suggesting a significant similarity between these two groups.

### 3.3. Reliability and Validity

We used four variables: burnout, homesickness, coworker homophily, and roommate homophily. All measurement items for the latent variables were assessed on a five-point Likert scale. Table 1 presents the results from reliability and convergent validity tests. As shown in Table 2, factor loadings for all items related to burnout and homesickness exceeded the recommended cutoff point of 0.70 ([34]). The factor loadings for two items each in coworker homophily and roommate homophily were slightly below this threshold. We retained these items for theoretical reasons in order to capture the full dimensions of these two variables.

Furthermore, Cronbach’s *α* and composite reliability scores for all four constructs surpassed the recommended cutoff point of 0.70 ([34]). The average variance extracted (AVE) value of each construct exceeded the commonly recommended cutoff value of 0.50 ([24]), suggesting satisfactory reliability and convergent validity.

We employed two methods for discriminant validity, with results reported in Table 3. First, the square root of the AVE (as shown in bold in the table) was higher than the highest correlation coefficient of the construct with any other variables in the analysis, thus meeting the criterion for discriminant validity defined by [24] ([24]), including for coworker homophily and roommate homophily. Second, we applied the heterotrait–monotrait ratio (HTMT), as developed by [38] ([38]), to assess discriminant validity. If the HTMT for all latent variables is below the predefined threshold of 0.85 ([50]), discriminant validity is established. As shown in Table 3, the discriminant validity of the measures was confirmed to be satisfactory, except for those of coworker homophily and roommate homophily, indicating some overlap between these two variables.

## 4. Results

We employed OLS regression analysis to test our hypotheses, with results reported in Table 4. Our baseline model in Column 1 demonstrates that homesickness was positively associated with burnout (β = 0.116, *p* < 0.01), supporting Hypothesis 1. This relationship remained robust across all model specifications.

The analysis of music listening during off-work time revealed two key findings. First, Column 2 shows that off-work music listening has a direct negative association with burnout (β = −0.107, *p* < 0.05). More importantly, Column 3 reveals a significant positive interaction between homesickness and off-work music listening (β = 0.210, *p* < 0.01), indicating that music listening strengthens the positive relationship between homesickness and burnout, as depicted in Figure 2. These findings support Hypothesis 2.

The most complex relationships emerge in the roommate-homophily analysis. Column 4 shows that roommate homophily is positively associated with burnout (β = 0.102, *p* < 0.01). More importantly, Column 5 reveals a significant three-way interaction between homesickness, off-work music listening, and roommate homophily (β = 0.197, *p* < 0.05). As illustrated in Figure 3, the moderating effect of music listening is particularly pronounced when roommate homophily is high. Specifically, the difference in the homesickness–burnout relationship between those who listen to music and those who do not is substantially larger under conditions of high roommate homophily compared to conditions of low roommate homophily. These findings provide strong support for Hypothesis 3a.

In columns 6 and 7, we replaced roommate homophily with coworker homophily and obtained similar results. Figure 4 illustrates this.

## 5. Discussion

This study examines the complex associations between homesickness, off-work music listening, and social homophily as predictors of burnout among migrant workers through the lens of the JD–R model and temporal comparison theory. Our findings unravel the interplays between these constructs, highlighting how these seemingly supportive resources can amplify psychological strain for migrant workers.

First, our finding on the positive association between homesickness and migrant worker burnout concurs with that of [92] ([92]) regarding how geographical separation creates psychological demands through rumination about home and attachment; it also aligns with that of [122] ([122]) on how nostalgia and separation can drain psychological resources. We are among the first to identify an important antecedent of burnout for migrant workers by extending this knowledge to them; for migrant workers, prolonged separation from familiar support systems can overwhelm adaptations ([59]).

Second, we find that listening to music during off-work time moderates the positive relationship between homesickness and burnout. This finding complements previous findings on the buffering effects of cultural resources ([45]). Our results indicate that music listening after work may lead to temporal comparisons that intensify rather than alleviate homesickness-caused burnout. This finding concurs with that of [29] ([29]) regarding how music can evoke strong emotional responses such as nostalgia and melancholy. Those responses can amplify feelings of loss and dissatisfaction ([29]).

Our findings on the role of homophily in amplifying homesickness-related strain resonate with theories of cultural capital and social distinction ([16]). While typically applied to elite populations, the mechanism of symbolic reinforcement through shared cultural practices is relevant to understanding migrant workers’ experiences. Homophilous networks serve to reinforce in-group boundaries and subjective distinction. Similarly, both roommate and coworker homophily among migrant workers may function as boundary-drawing mechanisms that intensify feelings of detachment from the host society and deepen awareness of what is ‘lost’ in migration. This perspective helps explain why seemingly supportive homophilic environments can paradoxically heighten feelings of displacement and burnout—they simultaneously affirm shared identity and emphasize distance from the host culture. The similarity lies not in the class position of the actors but in the relational function of homophily, which affirms in-group sameness and reinforces perceived out-group disaffiliation.

Third, our finding that coworker homophily strengthens the moderating effect of off-work music listening on the homesickness–burnout relationship highlights the dual role of coworker similarity in migrant workers’ psychological adjustment. On the one hand, shared migration motives, work attitudes, and cultural values can foster mutual understanding and emotional support ([21]). On the other hand, these similarities can result in co-rumination and excessive discussion of negative experiences that amplify their emotional impact ([33]; [73]). When migrant workers listen to music that evokes memories of home alongside culturally similar coworkers, their shared experiences and values may lead to deeper discussions about displacement and unmet aspirations, intensifying homesickness. This finding extends prior research by demonstrating how shared backgrounds can counter-intuitively amplify stress responses ([107]).

Similarly, we find that roommate homophily strengthens the moderating effect of off-work music listening on the homesickness–burnout relationship. This finding highlights the unique psychological dynamics of residential settings, where shared living spaces with individuals from similar backgrounds can create a potent context for collective reminiscence. Our findings on the amplifying effects of homophily align with those of emerging research on network structure and support functions. While homophilic networks provide emotional comfort through shared understanding, they may also facilitate recursive patterns of co-rumination that intensify rather than alleviate psychological strain. [101] ([101]) demonstrated that multiplex ties often provide more effective instrumental and cognitive support by offering alternative framings and disrupting recursive thoughts. For migrant workers, developing ties beyond those with similar backgrounds might provide access to new coping strategies and perspectives that could buffer against the amplifying effects we observed.

During off-work time, when migrant workers are likely to have cognitive bandwidth, they may be occupied by shared reflection on lost social networks and discontinued cultural practices triggered by music listening ([92]; [108]). For instance, roommates from adjacent hometowns may recognize common ceremonial melodies, enhancing the impact of such stimuli. Their similar ages and life stages may mean they share comparable memories of how these songs were experienced in their respective community experiences, potentially intensifying the contrast with their current circumstances ([10]).

The similarity in the moderating effects of coworker and roommate homophily suggests that the amplification mechanism operates through shared social identity rather than specific contextual factors. Both workplace and residential settings provide opportunities for collective temporal comparisons, wherein shared experiences and cultural triggers such as music raise awareness of separation and displacement. This extends the literature on acculturation stress ([121]) and social support networks ([107]) by demonstrating how shared backgrounds and experiences can intensify rather than alleviate stress responses.

Our findings also contribute to emerging approaches in quantitative cultural sociology that examine health-relevant practices with attention to homophily and identity performance. In a finding similar to that of [69] ([69]) regarding dietary practices among Italian youth, which revealed how self-imposed restrictions function as acts of identity-making within homophilic social circles, our study demonstrates how off-work music listening among migrant workers becomes a socially situated practice that reinforces group identity while potentially amplifying vulnerability. Our findings indicate that cultural practices related to well-being cannot be separated from the social meanings and affiliations through which they are enacted. We highlight the value of quantitative approaches that can capture these nuanced cultural dynamics.

### 5.1. Theoretical Implications

Our findings make several contributions to the study of the function of resources in contexts of geographical and cultural displacement among migrant workers. First, we extend the JD–R model by revealing how resources can paradoxically transform into demands for migrant workers. While the JD–R model traditionally posits that personal and social resources buffer against job demands ([4]), our findings demonstrate that familiar cultural elements such as music and social connections with similar others can amplify rather than mitigate strain. This transformation occurs when resources trigger upward temporal comparisons, raising awareness of the gap between migrant workers’ current circumstances and their idealized memories of home. Music that evokes nostalgic memories of one’s hometown may cause temporary comparison and intensify feelings of displacement ([29]). This explains why some supportive resources fail to buffer against burnout and proves that resource effectiveness depends not just on availability but on individuals’ psychological processes. We also extend the JD–R model’s assumption of resource stability and underscore the need to consider the dynamic interplay between resources and psychological demands among migrant workers.

Second, this study demonstrates how social contexts shape the collective comparison processes of migrant workers. While the conceptualization of temporal comparison theory focused on individual-level temporal comparisons ([1]), our findings reveal that when migrant workers with similar backgrounds listen to music together, their temporal comparisons may become mutually reinforcing, intensifying homesickness-related strain ([108]; [122]). This finding extends temporal comparison theory by illuminating the social mechanisms through which individual psychological processes can become collectively amplified.

Third, we integrate the JD–R model with temporal comparison theory to explain why and when resources may function as demands. Our findings demonstrate that the same resource (social similarity) can have varying effects depending on how it interacts with temporal-comparison processes in different settings. For instance, while coworker homophily may facilitate task-oriented support, it can also enable co-rumination about shared challenges, amplifying the psychological impacts of homesickness ([33]). Likewise, roommate homophily may intensify migrant workers’ reminiscences about their hometowns ([92]). This finding shows how contextual factors influence whether resources function as buffers or amplifiers of homesickness among migrant workers.

Fourth, we contribute to emerging perspectives in music and health research that challenge assumptions about the universally beneficial effects of music. While music is often promoted as a therapeutic resource, scholars increasingly recognise its potential to have both positive and negative psychological effects ([12]; [86]). Our research extends these insights by showing how music listening among migrant workers with their high-homophily roommates and coworkers can, under the specific psychosocial conditions of homesickness, amplify rather than relieve the strain of homesickness. By so doing, we also answer the recent calls to develop culturally sensitive frameworks for music-based interventions by considering the ambivalent effects of memory and group identity ([47]). For migrant workers, music interventions should take into account not only individual preferences but also social contexts and potential temporal-comparison effects.

### 5.2. Practical Implications

Our study has several practical implications. First, organisations should recognise the potential ambivalence of these resources rather than simply providing cultural resources or encouraging social connections. Instead, organisations should facilitate heterogeneous social interactions that bring migrant workers into contact with diverse individuals to mitigate the recursive patterns of homesickness-focused co-rumination. Organisations might also consider offering a variety of leisure activities beyond music listening, particularly those that focus on present engagement rather than triggering memories of hometowns.

Second, at the policy level, our findings suggest the need for more comprehensive support systems for migrant workers that address both psychological adjustment and social integration. Rather than assuming that cultural preservation and homophilic communities are universally beneficial, policymakers should consider how to balance between cultural continuity of homogenous migrant workers and the heterogeneity of their connections. This might include programs involving migrants from diverse backgrounds rather than primarily programs that preserve connections to home.

Third, music listening during off-work time can intensify rather than alleviate the homesickness–burnout relationship. These organisations should consider the timing and social context in which migrant workers consume music. For instance, collective music listening that evokes memories of one’s hometown may amplify feelings of displacement. Therefore, organisations could provide alternative stress-management techniques, such as mindfulness exercises or physical activities, that do not trigger temporal comparisons. Additionally, organisations might encourage workers to engage with music in ways that promote positive emotional regulation, such as listening to uplifting or neutral genres during working hours, while reserving nostalgic or culturally familiar music for moderated settings where its emotional impact can be managed.

Fourth, the similar effects of coworker and roommate homophily suggest that simply clustering workers from similar backgrounds, whether in work teams or living arrangements, may inadvertently amplify psychological strain. While shared cultural identity can provide emotional comfort, it can also facilitate co-rumination and collective reminiscence about home, intensifying feelings of displacement. Organisations should create balanced social environments that support cultural identity while encouraging new connections. For example, mixed-background work teams can foster the diversity of social connections and reduce the risk of co-rumination. At the same time, opportunities for cultural exchange, such as cultural festivals, can help workers celebrate their heritage without triggering excessive nostalgia.

Finally, we find that homesickness functions as a job demand rather than an adjustment challenge. Organisations should implement interventions and train managers to recognise signs of homesickness-related burnout and identify at-risk migrant workers early. Additionally, organisations could provide counselling services to address the challenges of maintaining psychological wellbeing while managing home ties.

### 5.3. Limitations and Future Research

We recognise limitations that can lead to future research directions. First, our cross-sectional design suggests that we may not have entirely captured the dynamic nature of how off-work music listening and social interactions shape temporal comparisons over time. While our findings provide valuable insights into the relationships between homesickness, music listening, and burnout, they do not account for how these processes may evolve across different stages of migration adjustment. Future research could employ experience sampling methods to examine how daily patterns of music listening and social interactions influence the homesickness–burnout relationship. This approach could reveal whether the amplification effects we observed vary across different times of day or different phases of migration, such as initial adjustment versus long-term adaptation.

Second, while we found that social homophily has amplification effects of across coworker and roommate contexts, future research could explore the underlying mechanisms more deeply. Qualitative methods such as semi-structured interviews could provide richer insights into how shared backgrounds influence the collective processing of homesickness through music. For instance, researchers could investigate how migrant workers discuss their experiences of displacement and cultural loss with peers from similar backgrounds and how these conversations are shaped by music listening. This could help explain why social similarity enhances temporal-comparison effects. Moreover, considering the differences between performative and participatory musical practices, studies could investigate whether social music activities, such as karaoke, that emphasize present-focused engagement might buffer rather than amplify homesickness. Such interventions could potentially foster [13]’s ([13]) reflective nostalgia (i.e., acknowledging loss while open to new possibilities), rather than restorative nostalgia idealising the past.

Third, homophily is a multidimensional concept. Current research has only examined similarities in seven aspects. Other dimensions of similarity, e.g., dialect language and ethnicity, may also matter. Future research could investigate further how these dimensions of homophily may play a role.

Fourth, we focused on rural-to-urban migrant workers in China. Future studies could test our framework in other migration contexts, such as in the contexts of international migrants and refugees. This could reveal how cultural distance and institutional factors moderate the relationships among cultural triggers, social similarity, and psychological strain. Comparing these contexts could provide a broader understanding of how migration-related challenges influence the psychological wellbeing of different populations.

Fifth, our focus has been on interpersonal and cultural dynamics. However, the emotional burden of homesickness and burnout among migrants cannot be fully understood without attending to the macro-social forces that structure migrant temporality and legal precarity. For instance, even emotional experiences such as burnout, homesickness, and the longing evoked by music are shaped by institutional architectures that govern time, mobility, and settlement ([48]). Future research could explore how broader frameworks of governance, labour management, and migration regimes shape the temporal horizons and psychological burdens of mobile labour populations.

## 6. Conclusions

This study unravels how seemingly supportive resources can paradoxically intensify burnout among migrant workers. Drawing on the JD–R model and temporal comparison theory, we reveal that off-work music consumption and social connections with coworkers and roommates can moderate the homesickness–burnout association. We highlight how cultural resources (i.e., off-work music listening) interact with temporal comparison and social resources. We unravel when and why cultural (i.e., music listening) and social resources (i.e., homophily) may help or hinder migrant workers’ adaptation. By doing so, we highlight the importance of context-specific strategies that address the psychological processes underlying resource utilization, ensuring that migrant workers can thrive in their new environments while maintaining meaningful connections to their cultural roots. While our study focuses on individual psychological processes and dyadic relationships, we acknowledge the importance of institutional and structural mediators that are not covered in this study. Future research could examine how employers’ dormitory-assignment policies and labour-management practices might reinforce or mitigate the psychosocial dynamics we observed. For instance, dormitory assignments based on hometown origin could inadvertently create environments that facilitate collective reminiscence and homesickness. Similarly, workplace policies regarding what types of music to play during breaks or shift structures might influence how workers engage with cultural resources in coping with homesickness and burnout.

## Figures and Tables

**Figure 1 behavsci-15-00666-f001:**
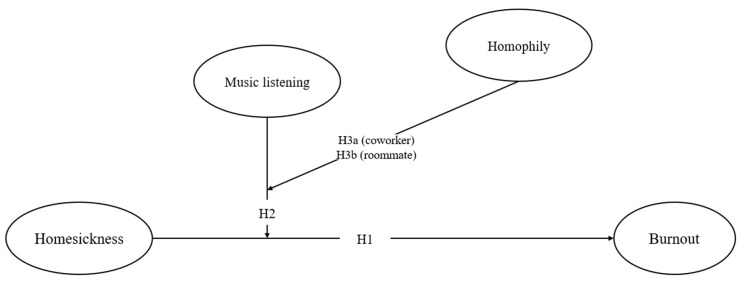
Conceptual model.

**Figure 2 behavsci-15-00666-f002:**
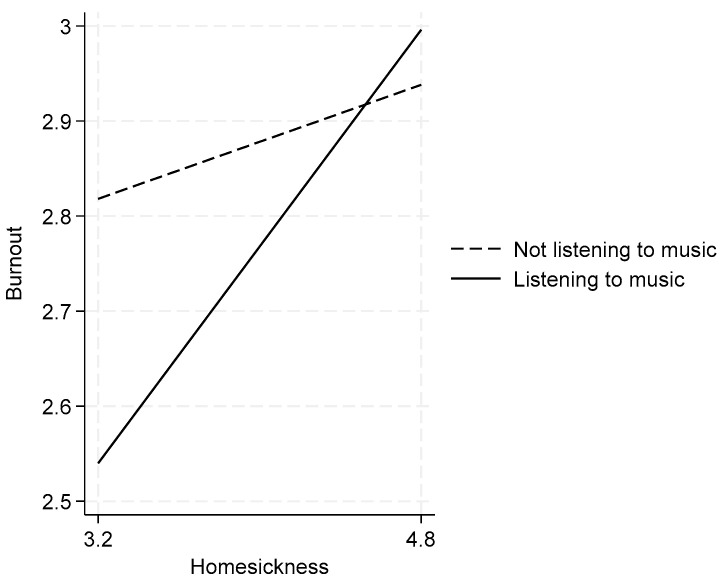
Homesickness and burnout: the moderating effect of off-work music listening.

**Figure 3 behavsci-15-00666-f003:**
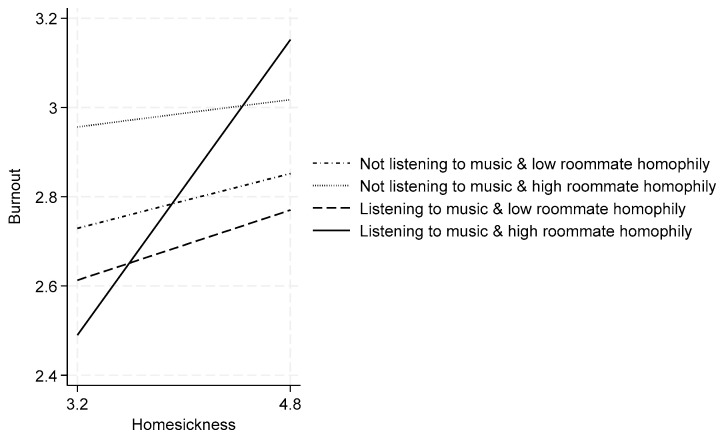
Homesickness and burnout: moderating effects of off-work music listening and roommate homophily.

**Figure 4 behavsci-15-00666-f004:**
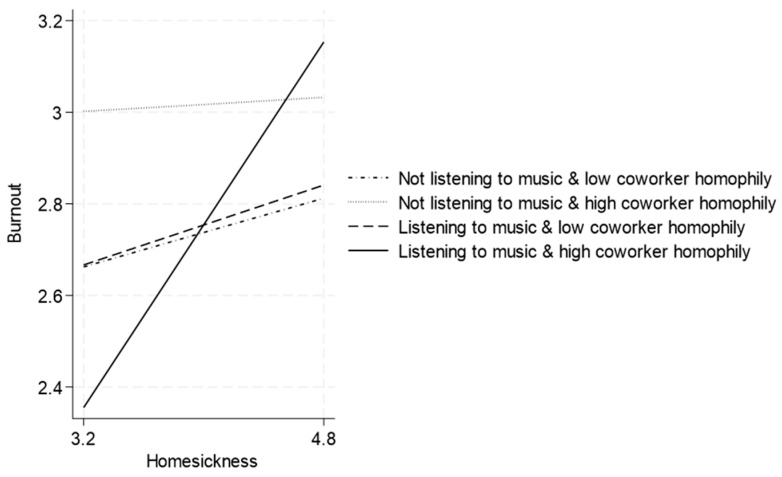
Homesickness and burnout: the moderating effects of off-work music listening and coworker homophily.

**Table 1 behavsci-15-00666-t001:** Descriptive statistics.

	Mean	Std. Deviation	1	2	3	4	5	6	7	8	9	10	11
Male	0.452	0.498	1										
Age	25.309	6.562	0.145	1									
Married	0.377	0.485	0.018	0.571	1								
Education	2.751	0.763	0.080	−0.041	−0.137	1							
Tenure	2.244	3.375	0.085	0.432	0.354	−0.051	1						
Cross-province	0.769	0.421	0.008	−0.006	0.024	−0.053	−0.022	1					
Burnout	2.871	1.128	−0.021	−0.103	−0.055	−0.035	−0.060	0.033	1				
Homesickness	4.007	0.817	−0.109	−0.015	0.036	−0.131	−0.012	0.066	0.074	1			
Off-work music listening	0.184	0.387	−0.039	−0.064	−0.070	0.044	−0.046	0.003	−0.049	−0.010	1		
Coworker homophily	3.233	0.852	−0.015	0.091	0.089	0.007	0.054	0.002	0.063	0.096	−0.003	1	
Roommate homophily	3.206	0.754	0.021	0.111	0.105	−0.004	0.087	−0.013	0.075	0.106	−0.040	0.697	1

**Table 2 behavsci-15-00666-t002:** Reliability and convergent validity.

Measurement Items	Factor Loading	Cronbach’s *α*	Composite Reliability	AVE
** Burnout (BU) **		0.817	0.894	0.738
BU1	0.811			
BU2	0.903			
BU3	0.860			
** Homesickness (HS) **		0.741	0.839	0.566
HS1	0.753			
HS2	0.728			
HS3	0.794			
HS4	0.732			
** Coworker homophily (CH) **		0.740	0.836	0.508
CH1	0.629			
CH2	0.749			
CH3	0.799			
CH4	0.783			
CH5	0.577			
** Roommate homophily (RH) **		0.803	0.867	0.568
RH1	0.697			
RH2	0.772			
RH3	0.810			
RH4	0.803			
RH5	0.676			

**Table 3 behavsci-15-00666-t003:** Discriminant validity.

Variable	Homesickness	Burnout	Coworker Homophily	Roommate Homophily
** Fornell-Larcker criteria **				
Homesickness	**0.752**			
Burnout	0.074	**0.859**		
Coworker homophily	0.096	0.063	**0.713**	
Roommate homophily	0.106	0.075	0.697	**0.754**
** Heterotrait-monotrait ratio (HTMT) **				
Homesickness				
Burnout	0.096			
Coworker homophily	0.165	0.094		
Roommate homophily	0.195	0.099	1.0196	

**Table 4 behavsci-15-00666-t004:** Results of hypothesis testing.

	(1)	(2)	(3)	(4)	(5)	(6)	(7)
Variables	Baseline	Music	Music	Roommates	Roommates	Coworkers	Coworkers
Male	0.0594	0.0560	0.0549	0.0682	0.0682	0.0653	0.0690
	(0.0467)	(0.0467)	(0.0467)	(0.0499)	(0.0499)	(0.0474)	(0.0473)
Age	−0.0168 ***	−0.0170 ***	−0.0168 ***	−0.0176 ***	−0.0176 ***	−0.0192 ***	−0.0190 ***
	(0.00469)	(0.00469)	(0.00469)	(0.00501)	(0.00501)	(0.00476)	(0.00475)
Married	−0.0364	−0.0399	−0.0408	−0.0642	−0.0608	−0.0230	−0.0187
	(0.0556)	(0.0556)	(0.0555)	(0.0601)	(0.0601)	(0.0562)	(0.0561)
Education	−0.0608 *	−0.0584 *	−0.0552 *	−0.0572	−0.0563	−0.0633 *	−0.0636 *
	(0.0326)	(0.0326)	(0.0325)	(0.0353)	(0.0353)	(0.0331)	(0.0330)
Tenure	−0.00299	−0.00314	−0.00284	−0.00634	−0.00688	−0.00606	−0.00688
	(0.00775)	(0.00774)	(0.00773)	(0.00856)	(0.00856)	(0.00789)	(0.00788)
Cross-province migrant	0.100 *	0.0980 *	0.0937 *	0.0796	0.0801	0.0914 *	0.0905 *
	(0.0530)	(0.0530)	(0.0529)	(0.0571)	(0.0571)	(0.0538)	(0.0537)
Homesickness	0.116 ***	0.116 ***	0.0750 **	0.0579 *	0.126	0.0595 *	0.205
	(0.0269)	(0.0268)	(0.0298)	(0.0324)	(0.122)	(0.0305)	(0.126)
Off-work music listening		−0.107 **	−0.950 ***	−0.941 ***	1.709	−0.898 ***	3.159 ***
		(0.0538)	(0.271)	(0.289)	(1.061)	(0.275)	(1.188)
Homesickness × Off-work music listening			0.210 ***	0.204 ***	−0.431 *	0.201 ***	−0.681 **
			(0.0662)	(0.0706)	(0.258)	(0.0672)	(0.289)
High-homophily roommates				0.102 ***	0.195		
				(0.0273)	(0.153)		
Homesickness × High-homophily roommates					−0.0214		
					(0.0369)		
Music off work × High-homophily roommates					−0.824 ***		
					(0.318)		
Homesickness × Off-work music listening× High-homophily roommates					0.197 **		
					(0.0768)		
High-homophily coworkers						0.143 ***	0.361 **
						(0.0296)	(0.160)
Homesickness × High-homophily coworkers							−0.0465
							(0.0385)
Music off work × High-homophily coworkers							−1.335 ***
							(0.376)
Homesickness × Off-work music listening × High-homophily coworkers							0.290 ***
							(0.0903)
Constant	2.913 ***	2.916 ***	2.809 ***	2.659 ***	2.368 ***	2.519 ***	1.845 ***
	(0.484)	(0.483)	(0.290)	(0.320)	(0.570)	(0.315)	(0.591)
Observations	2725	2725	2725	2377	2377	2625	2625
R-squared	0.089	0.090	0.094	0.101	0.104	0.105	0.110
Industry FE	YES	YES	YES	YES	YES	YES	YES

Note: Standard errors in parentheses; *** *p* < 0.01, ** *p* < 0.05, * *p* < 0.1.

## Data Availability

The original contributions presented in this study are included in the article. Further inquiries can be directed to the corresponding author.

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
