# Peer review of "When Cultural Resources Amplify Psychological Strain: Off-Work Music Listening, Homophily, and the Homesickness–Burnout Link Among Migrant Workers"

_behavsci, 2025, doi:10.3390/bs15050666_

Round 1

Reviewer 1 Report

Comments and Suggestions for Authors

This manuscript offers an important and highly original contribution to the interdisciplinary literature on migrant labor, psychological strain, and the ambivalence of social and cultural resources. Drawing on a large survey dataset of over 2,400 rural-to-urban migrant workers in Guangdong, China, the authors test the counterintuitive hypothesis that cultural resources—specifically, music listening in off-work settings—and social homophily may amplify, rather than alleviate, the psychological strain that emerges from homesickness. The study is situated at the intersection of the job-demands–resources (JD-R) model and temporal comparison theory and investigates the interactive effects of homesickness, music listening frequency, and homophilic proximity to coworkers or roommates on reported burnout. The originality of the argument, particularly in its critique of linear interpretations of the JD-R model, deserves further elaboration and alignment with contemporary developments in cultural sociology, public health musicology, and the sociology of elite reproduction.

This peer reviewer would like to express his admiration to your work and contribute in outlining how the manuscript could evolve into a more robust and theoretically expansive contribution.

The central strength of this manuscript lies in its theoretical originality and methodological rigor. The authors make a valuable intervention by challenging the commonly held assumption that cultural practices and affectively resonant social ties always function as protective buffers against stress and emotional vulnerability. Instead, they propose—and support through empirical modeling—that the nostalgic, identity-affirming dimensions of music and co-ethnic or co-class homophily may in fact enhance identity salience, evoke emotionally charged autobiographical memory, and exacerbate temporal comparisons between past and present, thereby intensifying the experience of burnout under conditions of displacement. The inclusion of a three-way interaction effect, and its graphical interpretation, is both technically sound and conceptually illuminating.

Nevertheless, the manuscript would benefit from several refinements—conceptual, theoretical, and empirical—that would deepen its sociological contributions and improve its resonance with a broader scholarly audience.

The first point of enhancement concerns the interpretation of “cultural resources” as inherently ambivalent. While the authors convincingly critique the linearity of the JD-R model’s treatment of resources as universally protective, they stop short of developing an adequate conceptual vocabulary for this ambivalence. To strengthen this argument, I suggest more explicit engagement with the literature on socially embedded coping, especially where it intersects with theories of co-rumination and emotional contagion in close-knit groups. This would clarify whether the amplification mechanism is primarily due to internal identity processing, as activated by music and memory, or social mirroring through repetitive, emotionally intense conversations within homophilic settings. These are distinct pathways, both theoretically and empirically, and distinguishing them could sharpen the causal narrative.

Secondly, while the JD-R model and temporal comparison theory offer a strong dual-theoretical base, the paper’s empirical findings call for an additional sociological lens: namely, the structure of personal networks and their influence on coping strategies. In this respect, I strongly recommend incorporating the work of Vacca et al. (2021) on non-ethnic, heterogeneous networks and their differentiated support functions. Vacca and colleagues argue that while homophilic networks may reinforce emotional salience, it is often heterophilous and multiplex ties that provide instrumental and cognitive forms of support, precisely by avoiding recursive co-rumination and offering alternative framings. Introducing this perspective in the discussion—particularly in the paragraph following line 445—would help the authors move from a binary logic of “homophily = risk” to a more refined understanding of support diversity, relational multiplexity, and tie heterogeneity. A few lines summarizing this literature would significantly broaden the manuscript’s impact and relevance.

Third, the paper currently lacks reflection on institutional or structural mediators. While the analysis focuses effectively on intra-individual and dyadic dimensions (i.e., music as self-regulation, homophily as social context), it leaves unexamined the potential role of organizational configurations, dormitory design, or labor management practices in either reinforcing or mitigating the psychosocial dynamics described. For instance, are workers assigned to dormitories based on origin or class background? Are musical listening practices spontaneous or mediated by shared access infrastructure (e.g., radio, communal Wi-Fi, curated playlists)? These questions do not necessarily require new data but could be introduced as speculative, interpretive avenues for future research, thereby enriching the paper’s structural sensitivity.

Moreover, the paper would benefit from a stronger methodological reflection on the meanings of “homophily” in the context of shared housing and labor environments. Currently, the operationalization of homophily as “similarity in work or housing characteristics” is analytically functional, but sociologically thin. I encourage the authors to reflect briefly on what kind of similarity matters: is it linguistic, ethnic, regional, class-based, or occupational? Adding even a few sentences in the methodology section or discussion acknowledging this multidimensionality—and the limitations of not having captured it—would help preempt interpretive overreach.

To continue, the ambivalence of cultural resources—especially when these are consumed in tightly homophilic contexts—could be productively enriched through the lens of cultural capital and social distinction, particularly as theorized in the work of Cousin and Chauvin (2017). In their study, they explore how social closure is maintained through ritualized cultural practices and homophilous consumption—music, food, social codes—which serve to reinforce in-group boundaries and subjective distinction. Although the manuscript focuses on migrant workers rather than elites, the mechanism of symbolic reinforcement through shared cultural practices bears strong analytical resonance. Off-work music listening, especially in group contexts marked by strong occupational or regional homogeneity, may similarly function as a boundary-drawing practice that intensifies feelings of detachment from the host society and deepens awareness of what is “lost” in migration. A short reflection—perhaps in the section following the discussion of social homophily—on the emotional reproduction of social difference through shared cultural consumption could open the manuscript to a broader debate on how class, identity, and memory interact in cultural practice. The similarity, here, lies not in the class position of the actors but in the relational function of cultural resources in affirming in-group sameness and reinforcing perceived out-group disaffiliation.

Another avenue for deepening the theoretical scope involves music and health scholarship, where the growing literature recognizes both the therapeutic and ambivalent potentials of music in contexts of psychological distress. Here, the contributions of Saarikallio (2012), Bonde and Theorell (2018), and Kang et al. (2025) are highly instructive. Saarikallio’s work highlights that while music can support mood regulation, it can also trigger negative affective states, especially when associated with unresolved autobiographical memories. Bonde and Theorell (2018), moreover, show that music interventions in public health must account for contextual and relational variables, not just musical content. This aligns with the manuscript’s central claim that music, far from being inherently healing, can act as a cognitive bridge to loss. Kang et al. (2025) go further, calling for a rethinking of music interventions to address health disparities in under-served populations, highlighting how class, culture, and spatial marginality affect the efficacy of musical health tools.

These contributions suggest that the manuscript should explicitly position its findings not only within organizational psychology or migration studies, but also within the public health discourse on culturally embedded health interventions. This could be done by inserting a short subsection or bridging paragraph in the discussion insisting that your findings support emerging insights in music and public health research that challenge the assumption of universal therapeutic benefit. As Kang et al. (2025) argue, music-based interventions require culturally sensitive frameworks that account for the ambivalent effects of memory and group identity. Similarly, Bonde and Theorell (2018) emphasize that musical experiences are not uniformly positive and depend on the context of listening and social framing. Youur study adds to this body of work by showing how informal music listening among migrant workers can, under specific psychosocial conditions, amplify rather than relieve strain.

This insertion would broaden the appeal of the paper to health promotion scholars, and show its alignment with critical perspectives on the social determinants of health. It would also allow the authors to link individual behavior (music listening) with wider systems of emotional regulation and social care, a theme central to current efforts to design community-based well-being interventions, such as those described in Heard and Bartleet’s (2025) study on music for collective well-being.

A third strand of integration should involve governance and state theory, especially through the lens of King, Le Galès, and Vitale (2017). Their work on assimilation and security regimes in contemporary European states points to the political structuring of migrant vulnerability and the state's role in the production of affective conditions such as anxiety, exclusion, and illegibility. While the manuscript does not explicitly theorize the role of the state, the emotional burden of homesickness and burnout among migrants cannot be fully understood without attending to the macro-social forces that structure migrant temporality and legal precarity. A short addition at the conclusion could acknowledge that while the paper focuses on proximate interpersonal and cultural dynamics, these dynamics occur within broader frameworks of governance, labor management, and disciplinary migration regimes, which shape the temporal horizons and psychological burdens of mobile labor populations. Quoting King, Le Galès et al. (2017), the authors could note that ven emotional life—burnout, homesickness, the longing evoked by music—is shaped by institutional architectures that govern time, mobility, and settlement. In this sense, your findings intersect with broader concerns about the governance of migrant subjectivity and the politics of affective inclusion.

Finally, in a different but complementary register, the manuscript would benefit from integrating recent advances in quantitative small-sample research on health and lifestyle homophily, especially in light of Morelli (2020) pivotal work. In his paper in Social Science Information on self-managed dairy restriction among Italian youth, Morelli show how dietary practices—framed as autonomous health choices—intersect with social differentiation and identity formation. Importantly, their work underscores how small-scale behavioral data can reveal deep connections between culture and health, particularly when behaviors are socially embedded and enacted collectively. Drawing a parallel, the authors of the current manuscript could explicitly situate their analysis as contributing to this emerging methodological and theoretical intersection: quantitative cultural sociology and health studies.

A paragraph near the conclusion could serve this purpose: your findings also resonate with emerging quantitative cultural sociology that examines health-relevant practices in small samples with fine-grained attention to homophily and identity performance. In their study of dairy restrictions among Italian youth, Morelli et al; (2020) reveal how self-imposed dietary constraints function as acts of identity-making, often performed within homophilic social circles. Analogously, you argue that off-work music listening among migrant workers, when shared with similar others, becomes a socially situated and affectively charged practice that can reinforce group identity and amplify vulnerability. This suggests that cultural practices related to health cannot be disentangled from the social meanings and affiliations through which they are enacted.

Such an addition would signal to the reader that the manuscript is in dialogue not only with occupational psychology but also with critical public health, cultural sociology, and the sociology of consumption.

In conclusion, this manuscript is already a compelling contribution. But with the integration of the literatures outlined above—on musical ambivalence, elite cultural reproduction, the governance of migrant affect, community-based music and health interventions, and small-sample cultural-health research—it could become a landmark paper. These suggested enrichments would not require the addition of new data or a shift in empirical orientation. Rather, they would extend the conceptual scaffolding of the work, enhancing its disciplinary permeability and strengthening its interpretive power. They would also anchor the manuscript in a broader effort to re-theorize cultural practices and social ties as relational, ambivalent, and structurally embedded mediators of health and well-being.

In the conclusion, the authors should also consider revisiting the normative implications of their findings. If off-work music and familiar social company can intensify burnout, how should organizations respond? Should they discourage communal music sharing, promote social heterogeneity, or invest in alternative well-being strategies? The authors wisely refrain from simplistic prescriptions, but a more explicit articulation of what this means for the design of support systems—especially in high-mobility labor contexts—would enhance the paper’s relevance for policymakers and HR professionals.

Finally, I wish to commend the authors for choosing to study off-work activities and informal relational contexts rather than formal workplace dynamics. This is a commendable shift that more labor and organizational psychologists should consider. However, to consolidate the theoretical coherence of the manuscript, the authors might consider introducing a short paragraph earlier in the introduction that situates this choice within the broader critique of psychological reductionism in burnout research, and affirms the value of cultural-sociological approaches to emotional life under precarity.

With these revisions, the manuscript will not only retain its empirical value but elevate its theoretical sophistication and interdisciplinarity. It has the potential to make a strong contribution to research at the intersection of migration studies, occupational health, and cultural sociology.

Author Response

Dear Reviewer, thank you for your comments. Please see the attachment.

Best wishes,

The Research Team

Reviewer 2 Report

Comments and Suggestions for Authors

I recommend to include the literature about music and diaspora (e.g. Tina Ramnarine (ed.), "Musical Performance in the Diaspora", Routledge: 2008), music and nostalgia (e.g. Eckehard Pistrick, "Performing Nostalgia: Migration Culture and Creativity in South Albania", Routledge: 2015), and please consider the concepts of reflective and restorative nostalgia (Svetlana Boym, "The Future of Nostalgia", 2001).

I suggest also to design and offer a model for social gathering with music to migrant workers, including not only performative, but also participatory event.

Author Response

(The authors gave the same response as above.)

Reviewer 3 Report

Comments and Suggestions for Authors

This article aims to investigate how off-work music listening, combined with social homophily (roommate and coworker homophily), interactively influence the homesickness-burnout link among migrant workers. The study integrates the job demands-resources model and temporal comparison theory.

3 hypotheses have been formulated based on literature review:

  • H1: Homesickness is positively related to burnout among migrant workers.
  • H2: Music listening during off-work time strengthens the positive relationship between homesickness and burnout among migrant workers.
  • H3a: Coworker homophily strengthens the moderating effect of non-working time music listening on the homesickness-burnout relationship, such that the positive relationship between homesickness and burnout becomes stronger when listening to music with high coworker homophily.
  • H3b: Roommate homophily strengthens the moderating effect of non-working time music listening on the homesickness-burnout relationship, such that the positive relationship between homesickness and burnout becomes stronger when listening to music with high roommate homophily.

The topic is very interesting and less presented in existing literature. The authors clearly identified the literature gap (lines 55-62): “homophily in social networks suggests people are mostly likely to attend cultural activities with people who share similar backgrounds”; “homophily in workplace and living environments can influence well-being and performance”; “music listening can affect employees’ psychological states”. Still “it remains unclear how music listening, combined with homophily, interactively influence the homesickness-burnout link among migrant workers”.  

This study makes several important theoretical and practical contributions which are very clear explained by the authors (chapters 5.1 and 5.2). The study extends the JD-R model and temporal comparison theory by revealing that familiar cultural elements (music and social connections with similar others) can amplify rather than mitigate the homesickness-burnout relation. At the same time, the authors highlight the implications for organizations, explaining “why some supportive resources (eg. collective musing listening, clustering workers from similar backgrounds, whether in work teams or living arrangements) fail to buffer against burnout and proves that resource effectiveness depends not just on availability but on individuals’ psychological processes.” 

The methodology is clearly described: the variables are well defined and the results of the hypothesis tests are clear.  The authors describe the limitations of the approach and the methodology. The conclusions address the main research objectives. At the same time, the authors highlight the theoretical and practical implications.

Recommendation:

The authors are using relevant literature references, pertaining to the investigated topic. However, it is recommended that:

  • in the Chapter 2. Theoretical background and hypothesis development, more updated references to be investigated and cited.
  • Lines 25-26: There are approximately one billion migrants worldwide, with 75% of them moving within their own country as internal migrants (Bell & Charles-Edwards, 2013). à is there a more recent resource on this information?

Author Response

(The authors gave the same response as above.)

Reviewer 4 Report

Comments and Suggestions for Authors

I consider this article a coherent and sophisticated examination of links between music listening and burnout among migrant worker in Guangdong, China. It gives evidence in support of important points, including one countering the typical view of off-work music listening as simply a “cultural resource” but instead suggesting how it might amplify the negative effects of homesickness. The paper also effectively integrates homophily into the equation to help explain the mechanism behind these effects, and makes clear suggestions as to the theoretical and practical applications of this knowledge. I also find the “Theoretical background and hypothesis development” section especially useful and rich in how it positions this study against multiple layers of existing discussions.

The paper certainly deserves to be published and here I just offer a few detailed comments that the authors might consider:

  • In the first paragraph of the introduction, it would be helpful to know how the authors are understanding the notions of “migrant” and “migrant worker” (lines 25-7) – some kind of working definition. I make this comment partly because these introductory sentences jump from talking about “migrants” (one billion worldwide) to “migrant workers,” which seems potentially to be two quite distinct concepts. Perhaps in line 25, the authors actually mean that there are one billion “migrant workers worldwide” – that would certainly make this passage clearer.
  • It would be helpful if the first time the JD-R model is referred to (line 34), the full name (“job demands-resources”) were written out in full. As it is, this does follow quite quickly, in the next paragraph, but it doesn’t seem to make sense to leave the reader in the dark for a while.
  • Line 40 and line 45-6: it is not very intuitive to me that homesickness be considered to fall within the “job demands” category. Perhaps overcoming or dealing with homesickness is something that “require[s] sustained physical or mental effort” in the process of pursuing one’s work; I think things only make sense if there is a verb attached to the noun homesickness. However the authors think about this (how they conceive of homesickness in relation to the categories of “job demands” and “job resources”), I think it would be useful if there were a few more words dedicated to explaining and justifying this crucial decision.
  • On line 56, it is unclear who “they” refers to in the construction “they are most likely to attend…”

Author Response

(The authors gave the same response as above.)
